# Nano- to macro-scale control of 3D printed materials via polymerization induced microphase separation

Valentin A. Bobrin[1], Yin Yao[2], Xiaobing Shi[1], Yuan Xiu[1], Jin Zhang ⬤ [3✉], Nathaniel Corrigan[1,4✉] & Cyrille Boyer ⬤ [1,4✉]

Although 3D printing allows the macroscopic structure of objects to be easily controlled, controlling the nanostructure of 3D printed materials has rarely been reported. Herein, we report an efficient and versatile process for fabricating 3D printed materials with controlled nanoscale structural features. This approach uses resins containing macromolecular chain transfer agents (macroCTAs) which microphase separate during the photoinduced 3D printing process to form nanostructured materials. By varying the chain length of the macroCTA, we demonstrate a high level of control over the microphase separation behavior, resulting in materials with controllable nanoscale sizes and morphologies. Importantly, the bulk mechanical properties of 3D printed objects are correlated with their morphologies; transitioning from discrete globular to interpenetrating domains results in a marked improvement in mechanical performance, which is ascribed to the increased interfacial interaction between soft and hard domains. Overall, the findings of this work enable the simplified production of materials with tightly controllable nanostructures for broad potential applications.

[1] Cluster for Advanced Macromolecular Design, School of Chemical Engineering, University of New South Wales, Sydney, NSW 2052, Australia. [2] Electron Microscope Unit, Mark Wainwright Analytical Centre, University of New South Wales, Sydney, NSW 2052, Australia. [3] School of Mechanical and Manufacturing Engineering, University of New South Wales, Sydney, NSW 2052, Australia. [4] Australian Centre for Nanomedicine, School of Chemical Engineering, University of New South Wales, Sydney, NSW 2052, Australia. ✉email: jin.zhang6@unsw.edu.au; n.corrigan@unsw.edu.au; cboyer@unsw.edu.au

Since its introduction in the 1980s, three-dimensional (3D) printing has revolutionized material synthesis. The development of extrusion, sintering, and vat photopolymerization based 3D printing has simplified the fabrication of materials with complex geometries and tailorable physico-chemical properties[1]. Indeed, functional polymer[2], metal[3], and ceramic materials[4] with intricate structures can be easily manufactured through various 3D printing techniques. Despite these advancements, the ability to provide structural control from the nano- to macroscale in 3D printed materials remains a great challenge, with only a handful of systems showing nanostructural control over 3D printed materials. Given the wide utility of nanostructured materials as cell-culture scaffolds[5], conducting materials[6], mechanical metamaterials[7], and energy device components[8], achieving precise control over the 3D printed material structures across multiple size scales could lead to new materials via simplified production routes.

To achieve nano- and microscale material structuration in 3D printing, there have been two main strategies explored in the literature. The first strategy relies on the development of precision hardware to decrease the voxel size in two-photon polymerization (2PP) processes[9–13]. These systems have successfully produced polymeric materials with controlled structures at sub-micrometer scale, which has led to materials with highly controlled chemical, biological, and optical properties. However, these systems suffer from low production rates and the need for precisely engineered and expensive equipment. The alternative to hardware-driven strategies are chemically controlled strategies for nano- and microscale structuration. Systems which use this strategy typically rely on phase separation between thermodynamically incompatible components to drive the formation of discrete nano- and microscale domains throughout the 3D printed material[14–18]. These chemical approaches are very noteworthy as they can produce nanostructured materials using inexpensive and highly accessible equipment and at higher throughputs.

As an example of this chemically mediated nanostructuration, Bates and co-workers 3D printed super-soft elastomers via a direct ink writing (DIW) technique by developing inks with bottlebrush copolymers which self-assembled into well-ordered body-centred cubic sphere (BCC) phases[17]. The length scale of microphase separation can be finely tuned by varying side-chain block length of bottlebrush copolymers. Importantly, a reversible structural transition between BCC lattice and disordered micelles that occurs in these bottlebrush self-assembled nanostructures under stress facilitates printability of the bottlebrush resin. In this case, the resulting 3D printed bottlebrush elastomers exhibited near-perfect recoverability well beyond the yield strain, however, the macroscopic material resolution was low due to the use of DIW 3D printing. Other groups[15,16,18] have applied polymerization induced phase separation (PIPS)[19,20] to photoinduced 3D printing to fabricate materials with micro-/nanoscale internal structures. The use of photoinduced techniques provides more highly resolved geometrical structures, however, the employed phase separation processes in these works resulted in limited control over nanostructuration.

Notably, our group recently developed a photoinduced 3D printing process using polymerization induced microphase separation (PIMS), to fabricate materials with nanostructured domains[21]. The PIMS process was originally developed by Seo and Hillmyer[22], and relies on in-situ chain extension of a macromolecular chain transfer agent (macroCTA) to induce microphase separation between incompatible block segments; the microphase separation is then arrested via crosslinking which provides materials with nanoscale morphologies (Fig. 1a). PIMS systems[23–27], including some photoinduced systems[28], allow a high degree of control over nanoscale morphologies and domain sizes. As a result of this exceptional nanoscale control, materials fabricated through PIMS processes have been used as polymer electrolyte membranes[29–33], heterogeneous catalysts[34], photochromic dye hosts[35], and nanostructured microneedles[36] and microcapsules[37] for tunable release of loaded molecules. Compared to these previous PIMS strategies, which were typically performed using temperatures above 100 °C and reaction times on the order of several hours, our previous work allowed 3D printing via PIMS to be performed in open-air conditions at room temperature with the reaction timescale of a few minutes. While 3D printing was successful, limited control over material nanostructuration was demonstrated with a lack of ability to finely tune nanoscale features (Fig. 1b).

In this work, we aim to bridge the gap between the high nanostructural tunability of PIMS and the simplified and versatile production methods of photoinduced 3D printing. Drawing inspiration from previous block polymer self-assembly and PIMS systems, we designed a series of macroCTAs with varied chain lengths and applied them to our vat photopolymerization based 3D printing system (Fig. 1c). By varying the macroCTA chain length and polymer volume fractions, structure-property relationships between the initial resin components and the resulting nanostructured morphologies were developed. A range of nanostructured morphologies were observed, from globular and discrete elongated domains to bicontinuous domains, and a scaling law was identified to describe the changes in nanostructure domain sizes that occur with increasing macroCTA chain length and volume fraction. Moreover, the dependence of the bulk mechanical properties on the nanoscale morphology of 3D printed materials was uncovered, where an optimal length scale was observed for bicontinuous nanostructured materials to exhibit enhanced mechanical properties. Finally, high-resolution macroscale materials with controlled internal nanostructures were 3D printed to demonstrate the versatility and robustness of our approach. As this work clearly outlines the co-dependence of the macroCTA degree of polymerization and the macroCTA concentration in determining the ultimate morphology, the design paradigms developed in this work should inform the future fabrication of hierarchically structured materials with medical, engineering, and energy applications.

## Results

**MacroCTA synthesis and polymerization kinetics**. To provide a range of macroCTAs with different chain lengths, we synthesized five PBA-CTAs via RAFT polymerization of $n$-butyl acrylate (BA) using 2-($n$-butylthiocarbonothioylthio) propanoic acid as RAFT agent and 2,2'-azobisisobutyronitrile as thermal initiator at 60 °C (Supplementary Fig. 1). The molecular weights of polymers were selected to be above and below the critical molecular weight for entanglement of linear poly($n$-butyl acrylate) (25 kg mol$^{-1}$)[38,39]. Specifically, the synthesized polymers had number-average molecular weights ($M_n$) in the range of 3.3–46.4 kg mol$^{-1}$ and low dispersities ($Đ < 1.19$) (Supplementary Fig. 2). The degree of polymerization ($X_n$) of the five PBA-CTAs as determined by proton nuclear magnetic resonance spectroscopy were 24, 48, 94, 180, and 360 (Supplementary Fig. 3). Further details on the polymer synthesis and characterization can be found in Supplementary Table 1. Altogether, well-defined PBA-CTAs were successfully prepared by RAFT polymerization, allowing us to investigate the effect of PBA-CTA chain length on model polymerization kinetics.

Subsequently, 15 resins were formulated by mixing AA, PEGDA, PBA-CTA, and TPO in predetermined weight ratios to form homogeneous, transparent mixtures (Supplementary Table 2). The photopolymerization kinetics of each resin was then

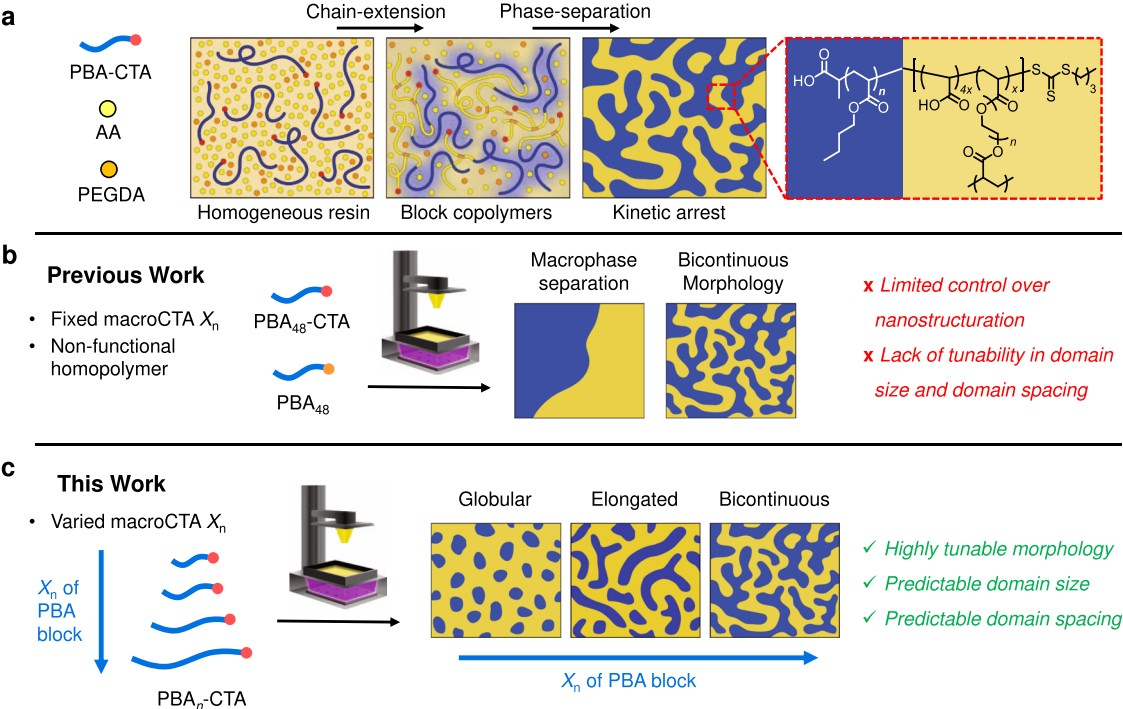

**Fig. 1 3D printing materials by RAFT-mediated PIMS. a** PIMS mechanism: a macroCTA is chain extended with acrylic acid (AA) and poly(ethylene glycol) diacrylate (PEGDA) to form block copolymers, which eventually phase-separate with the generation of emergent morphologies trapped by in-situ cross-linking. **b** Previous work using PBA-CTA with fixed chain length blended with non-functional PBA to 3D print PIMS materials with limited control over morphology. **c** Current work using PBA-CTAs with varied chain length to investigate structure-property relationships of 3D printed PIMS materials.

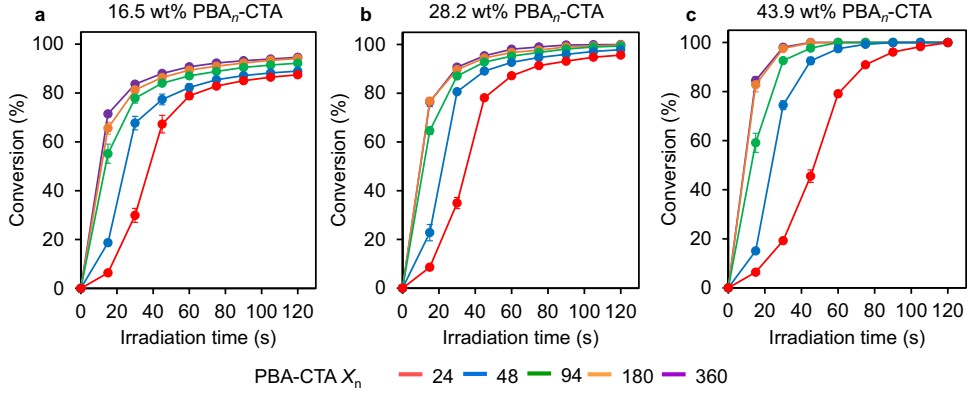

**Fig. 2 Effect of PBA-CTA chain length on polymerization kinetics.** The resins were formulated using a fixed molar ratio of [AA]/[PEGDA] = 4/1 and a varied weight percentage of PBA-CTA as follows: **a** 16.5 wt%; **b** 28.2 wt%; **c** 43.9 wt%. $X_n$ – PBA-CTA degree of polymerization. The kinetics experiments were performed in triplicate. Double bond conversions were determined using ATR-FTIR under 2.06 mW cm$^{-2}$ violet light ($\lambda_{max}$ = 405 nm). Error bars indicate standard deviation in triplicate measurements. Some error bars fall within the size of the markers.

determined in open-air conditions under 2.06 mW cm$^{-2}$ violet light irradiation ($\lambda_{max}$ = 405 nm) (Supplementary Fig. 4). The molar ratio of [AA]/[PEGDA] was fixed at 4/1 and the mass loading of PBA-CTA in the resins was varied between 16.5, 28.2, and 43.9 weight percent (wt%). The concentration of TPO was kept constant for resins with the same PBA-CTA wt%, specifically at 0.3, 0.5, and 0.7 wt% for 16.5, 28.2, and 43.9 wt% of PBA-CTA, respectively. The viscosity of resins increases upon increasing PBA-CTA $X_n$ and wt% of PBA-CTA (Supplementary Table 3 and Supplementary Note 2). All resin formulations demonstrated gelation within 60 s without a noticeable inhibition period, however, resins that contained macroCTAs with lower chain lengths ($X_n$ = 24 and 48) displayed slightly slower polymerization kinetics (Fig. 2). For example, the resin loaded with 16.5 wt% of

PBA$_{24}$-CTA showed 29.9% double bond conversion ($\alpha$) after 30 s of irradiation and $\alpha$ = 78.9% after 60 s (Fig. 2a). Comparatively, using PBA$_{94}$-CTA at 16.5 wt% loading resulted in $\alpha$ = 77.8% and 87.0% after 30 and 60 s of irradiation, respectively. Further increasing the macroCTA chain length to 180 and 360 repeating units of BA negligibly affected double bond conversion profiles, with these systems reaching high monomer conversions after 30 and 60 s with $\alpha \approx$ 84% and 91%, respectively. Similar trends in polymerization kinetics were observed for the 28.2 and 43.9 wt% systems (Fig. 2b, c). The slight reduction in polymerization rate for resins with low macroCTA chain lengths ($X_n$ = 24 and 48) can be explained by lower resin viscosity and the increased concentration of RAFT end-groups, in accordance with previous observations[40]. By fixing both the molar ratio of [AA]/[PEGDA]

and wt% of PBA-CTA, the number of polymeric chains containing RAFT groups increases as the PBA-CTA $X_n$ decreases (Supplementary Table 4). As the concentration of TPO was constant for resins with the same PBA-CTA wt% (Supplementary Table 2), systems with lower PBA-CTA $X_n$ displayed higher optical densities at 405 nm, and thus a lower photoinitiated radical generation rate (Supplementary Fig. 5 and Supplementary Note 3)[41]. Regardless, all resins demonstrated fast double bond conversion upon irradiation and appeared to be suitable for 3D printing.

**Morphology evolution in 3D printed PIMS materials**. To investigate the effect of PBA-CTA $X_n$ and wt% on the nanostructure of 3D printed materials, the 15 resins were applied to a commercial DLP 3D printer ($I_0 = 0.4$ mW cm$^{-2}$, $\lambda_{max} = 405$ nm) to fabricate model rectangular prisms ($l \times w \times t = 40 \times 8 \times 2$ mm). For comparison, the layer thickness and layer cure time for all the prints were set to 100 µm and 180 s/layer, respectively. All 3D printed materials were well-defined transparent rectangular prisms with high vinyl bond conversions ($\alpha > 91\%$) (Supplementary Table 2). After a 15 min post-cure treatment, the surface of 3D printed materials was examined by AFM to determine the formation of different microphase separated morphologies. PeakForce tapping mode was used to distinguish between domains with different mechanical properties, i.e., soft PBA domains (with a low modulus) and hard net-P(AA-stat-PEGDA) domains (with a high modulus). Representative AFM images are shown in Fig. 3, while lower magnification AFM images are presented in the Supplementary Figs. 6–8.

A clear morphological evolution was observed upon increasing the $X_n$ of the PBA block. For materials 3D printed with 16.5 wt%

of PBA-CTA and with $X_n = 24$ and 48, we observed the formation of discrete globular PBA domains dispersed in the continuous net-P(AA-stat-PEGDA) network (Fig. 3a, b). Further increasing the $X_n$ of the PBA block to 94 resulted in the formation of elongated PBA domains (Fig. 3c), while at $X_n = 180$ and 360, we observed bicontinuous morphologies (Fig. 3d, e). Close inspection of these AFM images revealed that the PBA domain width ($D_{PBA}$) and domain spacing ($d_{AFM}$) monotonically increased from 7 to 23 nm and from 19 to 55 nm, respectively, with increasing $X_n$ of the PBA block (Supplementary Figs. 10 and 11). Similar morphological evolutions were observed upon increasing PBA-CTA $X_n$ for materials 3D printed using resins with 28.2 and 43.9 wt% PBA-CTA. Materials with distinct globular PBA domains were observed when using PBA-CTA $X_n = 24$ (Fig. 3f, k), while elongated globular aggregates of PBA domains were observed in materials 3D printed using PBA-CTA $X_n = 48$ (Fig. 3g, l). When using resins with larger PBA-CTA chain lengths ($X_n = 94$, 180, and 360), materials with bicontinuous morphologies were obtained (Fig. 3h–j and Fig. 3m–o).

As an overall trend for materials 3D printed using the PIMS process, as the $X_n$ of PBA block increased from 24 to 360, $D_{PBA}$ and $d_{AFM}$ increased from 7 to 23 nm and from 15 to 55 nm, respectively (Supplementary Figs. 11–14). The increase in $D_{PBA}$ and $d_{AFM}$ in all cases was ascribed to the larger average block copolymer size prior to kinetic arrest, which resulted from the smaller number of RAFT capped chains at higher PBA-CTA $X_n$ (Supplementary Table 4). In addition, tan δ profiles for the objects 3D printed using 28.2 and 43.9 wt% PBA-CTA exhibited two separate peaks at around −35 and 75 °C associated with the glass transitions of the PBA-rich phase[42] and the net-P(AA-stat-PEGDA) phase[43], respectively, thus confirming the formation of microphase-separated morphologies (Supplementary Fig. 15 and

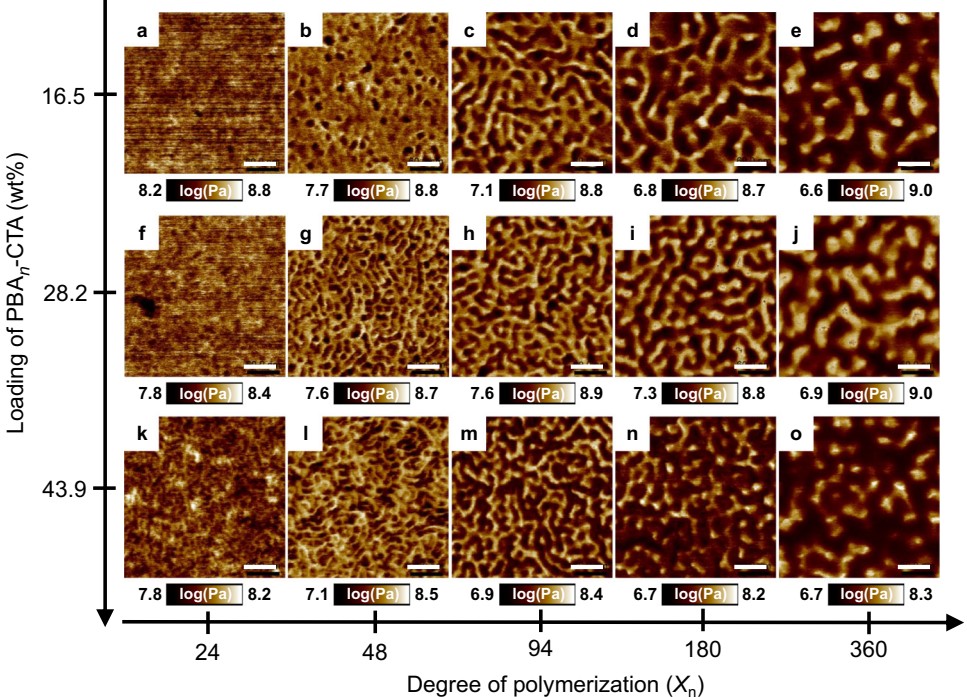

**Fig. 3 Dependence of PBA-CTA degree of polymerization ($X_n$) and PBA-CTA loading (wt%) on the surface morphologies of 3D printed materials.** **a–e** 16.5 wt% PBA-CTA with PBA-CTA $X_n = $ **a** 24, **b** 48, **c** 94, **d** 180, and **e** 360; **f–j** 28.2 wt% PBA-CTA with PBA-CTA $X_n = $ **f** 24, **g** 48, **h** 94, **i** 180, and **j** 360; **k–o** 43.9 wt% PBA-CTA with PBA-CTA $X_n = $ **k** 24, **l** 48, **m** 94, **n** 180, and **o** 360. PBA and net-P(AA-stat-PEGDA) domains are shown in dark brown and light brown, respectively. Materials were 3D printed using a molar ratio of [AA]/[PEGDA] = 4/1 at a fixed wt% of PBA-CTA and subsequently analyzed by AFM to obtain PeakForce QNM modulus map images. Scale bars are 60 nm. Note: Higher magnification of **a**, **f**, and **k** are shown in Supplementary Fig. 9.

Supplementary Note 4). These materials also demonstrated a drop in the storage modulus ($G'$) near $-40\,°C$ due to the glass transition of the PBA domains (Supplementary Fig. 16 and Supplementary Note 5). The gradual decrease in $G'$ continued with increasing temperature until the materials softened around $100\,°C$, likely due to passing through the glass transition of the net-P(AA-stat-PEGDA) domains (Supplementary Fig. 16).

Altogether, AFM results showed a clear morphological transition from globular to more continuous morphologies, i.e., elongated domains, and further to bicontinuous morphologies with increasing PBA-CTA $X_n$. In PIMS processes, the material morphology is dictated by competition between the thermo-dynamic forces driving microphase separation of the chemically incompatible blocks and the kinetic arrest of the network due to the in-situ crosslinking and gelation[25,44]. As a measure of the thermodynamic driving force, the segregation strength $\chi N$ is used, where $\chi$ is the interaction parameter and $N$ is the total degree of polymerization[45]. For PBA-b-(P(AA-stat-PEGDA)) block copolymers, $\chi$ was estimated to be 0.505 at 25 °C (Supplementary Note 1), indicating that the PBA and P(AA-stat-PEGDA) are thermodynamically incompatible[46]. Upon fixing the weight loading of PBA-CTA, the molar ratio of [AA]/[PEGDA], and the monomer conversion, increasing PBA-CTA $X_n$ results in a lower number of RAFT capped chains (Supplementary Table 4). Correspondingly, for resins with higher PBA-CTA $X_n$, the average PBA-b-(P(AA-stat-PEGDA)) block copolymer size prior to kinetic arrest is larger, as is $\chi N$ (Supplementary Table 4). While the polymerization rate increases at higher $N$, resulting in a shorter time to gelation (Fig. 2), the much higher $\chi N$ allows the formation of more developed, i.e., more continuous, and coarser, morphologies prior to kinetic arrest.

To further demonstrate that the $\chi N$, rather than the polymerization rate, had a greater impact on nano-scale morphology formation, 3D printing was performed using resins with different PBA-CTA $X_n$, but closely matching kinetic profiles. PBA-CTAs with $X_n = 24$ and 360 were selected for this comparison in a system with 16.5 wt%, due to their different morphologies and polymerization rates from the initial investigation (Fig. 2a and Fig. 3a, e). To reduce the polymerization rate of the resin with 16.5 wt% PBA-CTA $X_n = 360$, the concentration of TPO was reduced from 0.3 to 0.133 wt%, which provided kinetic profiles that were well-aligned with those of the 16.5 wt% PBA-CTA $X_n = 24$ system (Supplementary Fig. 17a). The formation of globular PBA domains was observed for a material 3D printed with PBA$_{24}$-CTA, while a tougher material with bicontinuous morphology was obtained with PBA$_{360}$-CTA (Supplementary Fig. 17b–d). Notably, the domain spacings of the materials printed using the 16.5 wt% PBA-CTA $X_n = 360$ resins at both polymerization rates were very similar, as visualized by AFM (Supplementary Fig. 6e, Fig. 11 and Fig. 17d–f). This result indicates that the morphology development of these 3D printed materials is primarily governed by the $\chi N$ of the block copolymers.

To gain further insight into the internal nanostructure of our 3D printed materials, SAXS experiments were performed. The typical sample for SAXS was a square prism with dimensions $l \times w \times t = 6 \times 6 \times 0.2\ mm$, 3D printed using $2 \times 100\ \mu m$ layers. Figure 4a–c represent the SAXS profiles of materials 3D printed using PBA-CTAs with varying $X_n$ from 24 to 360 at three different loadings of PBA-CTA, i.e., 16.5, 28.2 and 43.9 wt%. A single broad scattering peak was observed for all samples, indicating a disordered, microphase-separated state in the PBA-b-(net-P(AA-stat-PEGDA)) block copolymer network. It should be noted that higher-order peaks were not observed in SAXS patterns, which is consistent with previous reports on PIMS[22,25], and indicates lack of periodic order. For all three loadings, the

scattering peak maximum position shifted to lower $q$ upon increasing $X_n$, indicating that materials 3D printed using longer PBA-CTA chain lengths showed larger domain spacing. This observation is consistent with previous reports of microphase-separated cross-linked monoliths prepared by PIMS using macroCTAs with different molar masses[25,44]. The domain spacing calculated from SAXS ($d_{SAXS}$) increased with higher PBA-CTA $X_n$, which was attributed to the increased average block copolymer size prior to kinetic arrest (Supplementary Table 4). Notably, the $d_{SAXS}$ values were in close agreement with the domain spacing determined from AFM ($d_{AFM}$), which supports the morphologies obtained by AFM (Supplementary Table 5).

To compare the peak breadth, SAXS scattering plots were normalized based on the position ($q^*$) and intensity ($I^*$) of the principal peaks (Fig. 4d–f). For all three loadings of PBA-CTA, the breadth of SAXS peaks increased with higher PBA-CTA $X_n$, suggesting that materials 3D printed using higher PBA-CTA $X_n$ consisted of domains with less sharp compositional interfaces. To further examine this, the SAXS curves were fitted using the Teubner-Strey (T-S) model[47–49], which has been broadly applied for structural analysis of phase-separated polymers[50,51], including polymer materials prepared by PIMS approaches[44]. Three structural parameters are determined from the T-S model fitting: the domain spacing ($d_{TS}$), the correlation length ($\xi$), and the amphiphilicity factor ($f_a$). $d_{TS}$ describes the periodic spacing between domains, $\xi$ reflects the spatial coherence of the interfaces, and $f_a$ characterizes the segregation strength at the interfaces. When $f_a > 0$, the material is weakly-structured, $-1 < f_a < 0$ corresponds to well-structured materials, and $f_a = -1$ corresponds to interfacial segregation strengths for structures in ordered lamellar morphologies. The fitted SAXS patterns and extracted T-S model parameters are presented in Supplementary Table 6 and Supplementary Figs. 18–20. It should be noted that domain spacing values ($d_{TS}$) determined from fitting were in close agreement with $d_{SAXS}$ values (Supplementary Table 6). Importantly, the ratio of $\xi/d_{TS}$, which is a measure of the domain size polydispersity[52], decreased with increasing PBA-CTA $X_n$, indicating that the polydispersity of the domain size increased (Supplementary Table 6). In addition, for all samples, $f_a$ was found to be in the range from $-0.77$ to $-0.93$, suggesting 3D printed PIMS materials with well-structured domains and sharp interfaces. Essentially, $f_a$ values were close to $-0.9$ for PBA-CTA $X_n = 24$, 48, 94 and 180, and around $-0.8$ for PBA$_{360}$-CTA (Supplementary Fig. 21), which aligns with the SAXS peak broadening (Fig. 4d–f). This indicates the formation of domains with less sharp interfaces and broader polydispersity, which likely results from lower mobility of chains at PBA-CTA $X_n = 360$ due to the high molecular weight ($M_n = 46.4\ kg\ mol^{-1}$) which exceeds the critical molecular weight for entanglement ($25\ kg\ mol^{-1}$)[38,39].

Importantly, the ability to tune the domain spacings of microphase-separated 3D printed materials provides greater control over the resulting material properties[25]. It has been observed in other microphase-separated structures[45] that the domain spacing is proportional to the degree of polymerization, following the power law $d \sim N^a$. To demonstrate that the nanostructure of the 3D printed PIMS materials follows a predictable trend, $d_{SAXS}$ as function of the total degree of polymerization ($N'_{total}$) was fitted with a power law model (Fig. 5). The obtained scaling exponents were similar for all three loadings of PBA-CTA, giving $d_{SAXS} \sim N'^{3/5}_{total}$, which is consistent with PIMS literature[44]. From theory of microphase-separated block copolymers[45], in the weak segregation limit $d \sim N^{1/2}$ and polymer chains exist in unperturbed (i.e., Gaussian) conformation, while in the strong segregation limit (SSL) the polymer chains adopt perturbated, more stretched configuration with $d \sim N^{2/3}$. For our

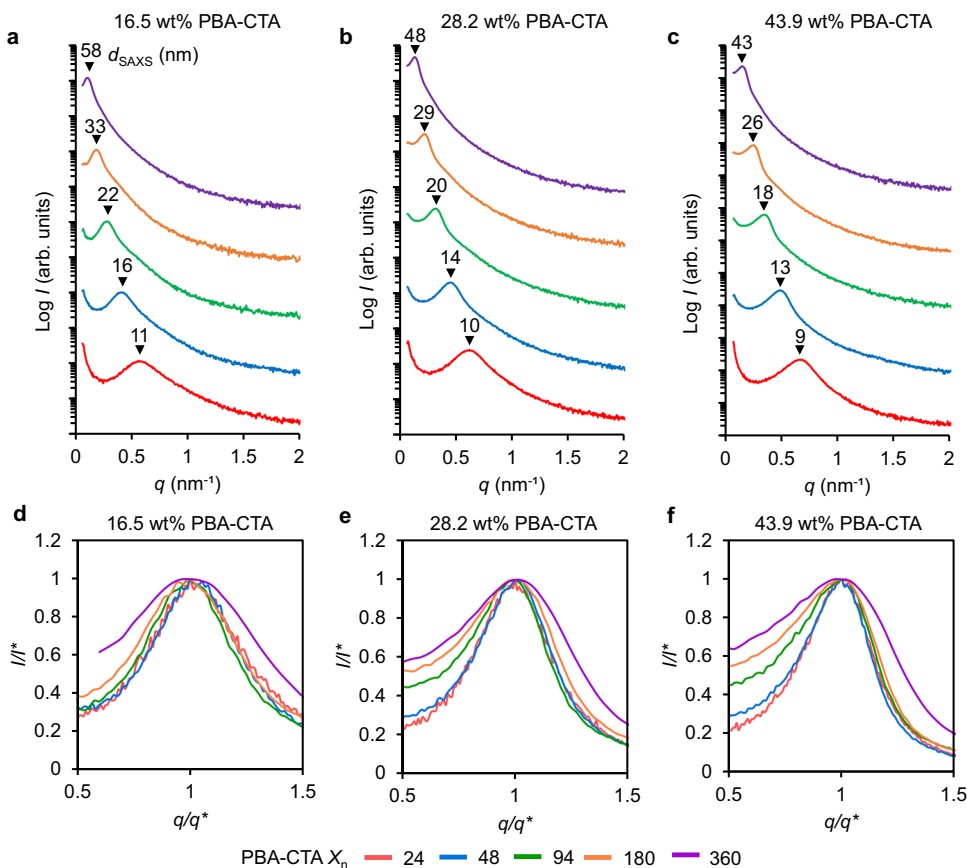

**Fig. 4 SAXS data for materials 3D printed using PIMS resins. a–c** SAXS profiles and corresponding domain spacing ($d_{SAXS}$) values of materials 3D printed using varied PBA-CTA wt%: **a** 16.5 wt% PBA-CTA; **b** 28.2 wt% PBA-CTA; **c** 43.9 wt% PBA-CTA. SAXS profiles were shifted vertically for clarity. **d–f** Scaled SAXS data showing comparison of peak broadness for materials 3D printed with various degree of polymerization ($X_n$) and PBA-CTA wt%: **d** 16.5 wt% PBA-CTA; **e** 28.2 wt% PBA-CTA; **f** 43.9 wt% PBA-CTA. For the data shown in d-f, the SAXS spectra were normalized based on the intensity ($I^*$) and the position ($q^*$) of the principal peaks.

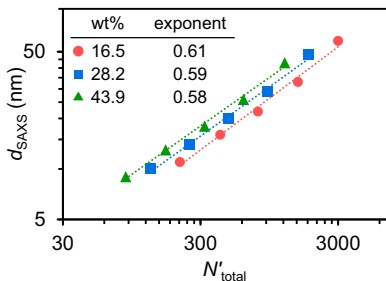

**Fig. 5 Power law scaling for domain spacings of 3D printed PIMS materials.** Log-log plot of domain spacing ($d_{SAXS}$) as a function of total degree of polymerization ($N'_{total}$). $d_{SAXS}$ determined from SAXS. $N'_{total}$ was calculated based on a common monomer reference volume (118 Å³). Power law regression lines and scaling exponents (inset table) are shown.

block copolymer system, $\chi N \gg 10$ (Supplementary Table 4), which indicates SSL[45]. However, the determined scaling exponent ($\delta$) of 3/5 is lower than the one reported for SSL ($\delta = 2/3$), suggesting that in our system polymer chains are less stretched compared to SSL regime. This is ascribed to the in-situ cross-linking during PIMS, which arrests polymer chains before they adopt stretched conformations typically observed in the SSL. Overall, the obtained scaling law provides a method for predicting nanoscale feature sizes of 3D printed PIMS materials based on experimentally tunable parameters.

**Mechanical behavior of 3D printed PIMS materials**. Having demonstrated the effect of PBA-CTA chain length on nanostructured material morphologies, the bulk mechanical properties of the 3D printed materials were examined (Supplementary Table 5 and Supplementary Fig. 22). As shown in Fig. 6a, upon increasing PBA-CTA $X_n$ from 24 to 94, the materials 3D printed with 28.2 and 43.9 wt% of PBA-CTA demonstrated similar values of stress at break ($\sigma_B$), while the material 3D printed with 16.5 wt% of PBA-CTA showed increased $\sigma_B$ from 38.7 to 48.1 MPa. Within the same range of PBA-CTA $X_n$, the elongation at break ($\varepsilon_B$) for all materials increased, with the highest increase from 40.7 to 91.4% observed for the material 3D printed with 16.5 wt% PBA-CTA (Fig. 6b). Consequently, the toughness of this material increased by nearly three-fold, from 13.3 to 35.5 MJ m⁻³, whereas materials 3D printed with higher loadings of PBA-CTA (28.2 and 43.9 wt%) exhibited slightly increased toughness (Fig. 6c). The improvement in material mechanical properties, particularly notable for materials 3D printed with a lower loading of PBA-CTA (16.5 wt%), can be attributed to the morphology transition from discrete globular PBA nanodomains dispersed in the *net*-P(AA-*stat*-PEGDA) matrix, to more continuous, interpenetrating soft PBA and hard *net*-P(AA-*stat*-PEGDA) nanodomains, thus allowing a more even distribution of stress throughout the material, which is consistent with the previous reports[42,53–55]. Fractography studies using SEM also revealed that 3D printed materials with interpenetrating morphologies, i.e., elongated PBA domains and bicontinuous structure, had rougher fracture surfaces compared to materials with globular morphologies, which

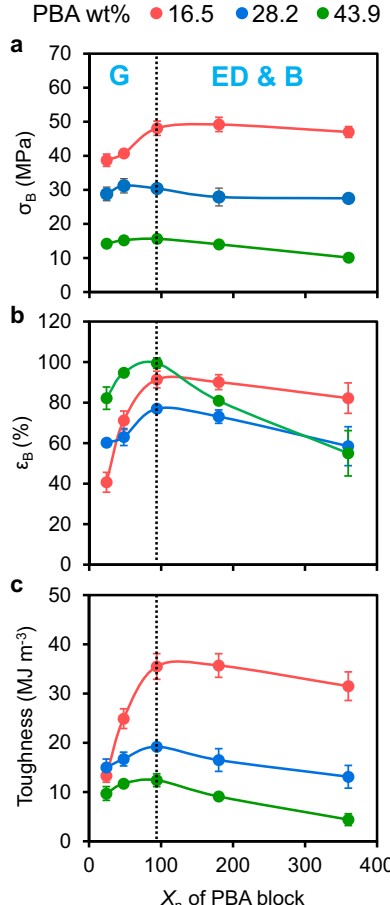

**Fig. 6 Bulk mechanical (tensile) properties of samples 3D printed with varying PBA-CTA $X_n$ and PBA-CTA wt%. a** Tensile stress at break ($\sigma_B$); **b** Elongation at break ($\varepsilon_B$); **c** Toughness. Materials were 3D printed using a molar ratio of [AA]/[PEGDA] = 4/1 at a fixed PBA-CTA wt% of (16.5, red lines, 28.2, blue lines, or 43.9 wt%, green lines). G - globular morphology, ED - elongated domains, B - bicontinuous morphology. Dashed line represents the boundary between globular (G) and interpenetrating morphologies (ED and B). Error bars indicate standard deviation in at least triplicate measurements. Some error bars fall within the size of the markers.

aligned with the increased fracture energy absorption and improve in toughness (Supplementary Figs. 23–25 and Supplementary Note 6)[56].

Further increasing the PBA-CTA $X_n$ from 94 to 360 resulted in an overall reduction in mechanical properties, with the higher PBA-CTA wt% materials showing a more pronounced decrease. For example, while the 16.5 wt% PBA-CTA system showed only marginal decreases in $\sigma_B$ from 48.1 to 47.0 MPa upon increasing $X_n$ from 94 to 360, the 43.9 wt% system showed a larger decrease from 15.6 to 10.1 MPa. The $\varepsilon_B$ also dramatically decreased for the 43.9 wt% PBA-CTA material, from 99.5% for the $PBA_{94}$-CTA material, to 55.0% $PBA_{360}$-CTA material (Fig. 6a, b). Consequently, the material toughness decreased more significantly for the 43.9 wt% system compared to the 28.2 and 16.5 wt% systems (Fig. 6c). SEM images showed smoother fracture surfaces for materials 3D printed at higher $X_n$, which indicates a decrease in toughness[56] (Supplementary Figs. 23–25 and Supplementary Note 6). We postulated that the mechanical properties changes that occurred with changing $X_n$ and PBA-CTA wt% were related to the domain size variations. To more closely examine this, $\sigma_B$, $\varepsilon_B$ and toughness were plotted as functions of $D_{PBA}$ and domain $d_{SAXS}$ for two types of interpenetrating morphologies, i.e.,

elongated domains and bicontinuous morphologies (Supplementary Fig. 26).

The decrease in the mechanical properties of the 3D printed materials with bicontinuous morphologies correlated with an increase in $d_{SAXS}$. For example, the material 3D printed with 43.9 wt% of $PBA_{94}$-CTA exhibited a bicontinuous morphology with $d_{SAXS} = 18$ nm; $\sigma_B$, $\varepsilon_B$, and toughness of this material were 15.6 MPa, 99.5% and 12.4 MJ m$^{-3}$, respectively. As the $d_{SAXS}$ values increased to 43 nm for the bicontinuous material 3D printed using 43.9 wt% of $PBA_{360}$-CTA, $\sigma_B$ decreased to 10.1 MPa while $\varepsilon_B$ and toughness decreased to 55.0% and 4.4 MJ m$^{-3}$, respectively (Supplementary Fig. 26a–c). The same correlation of material mechanical properties with $D_{PBA}$ was observed (Supplementary Fig. 26d–f). These results agree with literature precedents[42,53] and can be explained as follows: at a fixed weight loading of PBA-CTA, an increase in domain size and domain spacing reduces the interfacial area between soft and hard domains. This consequently reduces the efficiency of localized stress dissipation from hard to soft domains and lowers the amount of absorbed deformation energy required to cause fracture of a material.

Altogether, the mechanical properties of 3D printed materials with interpenetrating soft and hard domains, i.e., elongated domains and bicontinuous morphology, with length scales of $D_{PBA}$ ~13 nm and $d_{SAXS}$ ~20 nm were higher than similar materials with globular morphologies. However, further increasing in the $d_{SAXS}$ of bicontinuous materials resulted in reduced mechanical properties due to reduced interaction between soft and hard domains and lower dissipation of localized stress throughout the material. It is worth noting that the sharpness of interphase between PBA and *net*-P(AA-*stat*-PEGDA) domains, as quantified by $f_a$ parameter, may impact the toughness of 3D printed PIMS materials. To investigate this, two PIMS materials 3D printed using the same macroCTA of $PBA_{360}$-CTA at 16.5 wt % loading, but with different TPO wt%, 0.3 and 0.133 wt%, were compared. Both materials feature bicontinuous morphology with very similar values of $D_{PBA}$ and $d_{SAXS}$ (Supplementary Fig. 27a, b). Fitting SAXS data using the T-S model revealed different $f_a$ values: $f_a = -0.77$ and $-0.68$ for the materials 3D printed with 0.3 and 0.133 wt% TPO, respectively. This result suggests the formation of domains with more diffuse interface for material 3D printed with lower TPO wt%, which may result from the difference in the polymerization rate (Supplementary Fig. 27c). Consequently, the material 3D printed with higher TPO wt% exhibited improved mechanical properties: $\sigma_B$ and $\varepsilon_B$ increased from 39.2 to 47 MPa and from 63% to 82.2%, respectively, resulting in the increase in toughness from 20.3 to 31.5 MJ m$^{-3}$ (Supplementary Table 7).

**Macroscale geometric control via 3D printing.** To demonstrate the capability of RAFT-mediated PIMS 3D printing to fabricate complex objects that are challenging to produce via traditional manufacturing approaches, a cubic lattice structure with target strut width of 0.9 mm was designed and 3D printed (Supplementary Fig. 28). 3 PIMS resins were formulated with a molar ratio of [AA]/[PEGDA] = 4/1 and 28.2 wt% of PBA-CTA with $X_n = 48$, 94 and 180. The layer cure time was set to 25 s/layer and the layer thickness was 100 μm, which represents a reasonably practical build rate of 1.44 cm h$^{-1}$. As shown in Fig. 7a–c, for all formulated resins the 3D printed cubic lattice structures replicated the original CAD model (Fig. 7d) with high printing fidelity. The measured strut width of the 3D printed lattice was 0.8 mm (Fig. 7e) as measured by digital calipers, which is slightly lower than the target value of 0.9 mm. This is due to volume layer shrinkage commonly observed for acrylate resins[2]. Interestingly,

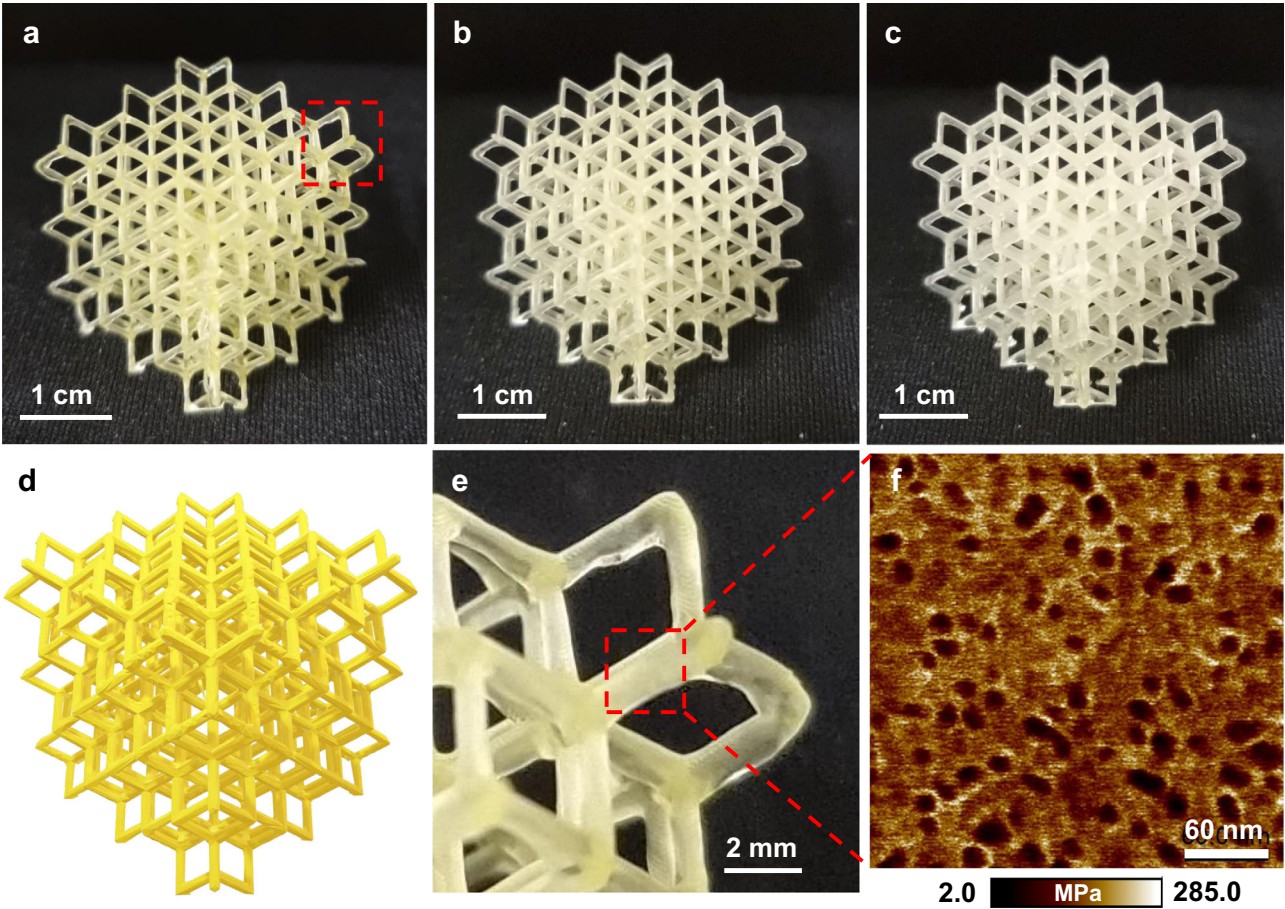

**Fig. 7 3D print of a complex structure.** Cubic lattice structures were 3D printed using PBA-CTAs with various degree of polymerization ($X_n$): **a** $X_n = 48$; **b** $X_n = 94$; **c** $X_n = 180$; **d** computer aided design of lattice structure to be 3D printed; **e** Close-up view and **f** surface morphology of 3D printed strut-based lattice structure using PBA$_{48}$-CTA. Materials were 3D printed using a molar ratio of [AA]/[PEGDA] = 4/1 at 28.2 wt% loading of PBA-CTA.

the tone of the 3D printed lattice structures became lighter upon increasing PBA-CTA $X_n$ from 48 to 180, which was caused by the lower concentration of polymeric chains containing trithiocarbonate groups (Supplementary Table 4).

Surface analysis of the lattice structure 3D printed using PBA$_{48}$-CTA revealed the formation of PBA globular aggregates with $D_{PBA} = 13$ nm and $d_{AFM} = 20$ nm dispersed in *net*-P(AA-*stat*-PEGDA) matrix (Fig. 7f and Supplementary Fig. 29). For lattice structures 3D printed with PBA-CTA $X_n = 94$ and 180, we observed elongated globular aggregates of PBA domains; $D_{PBA}$ and $d_{AFM}$ increased from 15 to 21 nm and from 25 to 33 nm, respectively, as PBA-CTA $X_n$ increased from 94 to 180 (Supplementary Fig. 30). Altogether, these results demonstrated effective fabrication of complex objects with precisely controlled macro- and nanostructures and high structural integrity using RAFT-mediated PIMS 3D printing. Furthermore, to demonstrate the impact of material nanostructuration on mechanical properties of 3D printed lattices, resins with and without PBA-CTA were selected to 3D print body-centered cubic lattice structures (Supplementary Fig. 31). Uniaxial compression testing was conducted to obtain the stress-strain curves of PIMS and non-PIMS lattice structures (Supplementary Fig. 32). Notably, the mechanical properties of materials 3D printed using PIMS resins were significantly higher than non-PIMS materials (Supplementary Table 8). Specifically, the Young's modulus, modulus of resilience, and toughness increased by 4-, 10-, and 3-fold, respectively, compared to non-PIMS counterparts. These results

highlight the beneficial effect of nanostructuration to enhance mechanical properties of 3D printed materials.

In conclusion, we investigated the effects of varying macroCTA chain length and polymer weight fractions on the resulting nanostructures of materials 3D printed by RAFT-mediated PIMS. We demonstrated an efficient method to 3D print materials with well-defined nanostructures including globular, elongated domains, and bicontinuous morphologies. The domain size and domain spacing of these microphase-separated nanostructures were largely defined by the degree of polymerization of the PBA macroCTA block and the PBA-*b*-(P(AA-*stat*-PEGDA)) block copolymer, respectively. More importantly, the domain sizes of these nanostructured materials followed predictable scaling behavior, increasing according to a power law with increasing block copolymer size. As such, the nanostructures of these 3D printed materials can be predetermined.

In addition, structure-property relationships for the nanostructured 3D printed materials were determined and revealed that the bulk mechanical properties were improved for materials with interpenetrating morphologies compared to those with globular structures. The increase in mechanical properties was ascribed to the increased interfacial interaction between soft and hard domains, which led to more efficient stress dissipation within the material. For materials with bicontinuous morphologies, an optimized length scale to obtain the highest mechanical properties was observed; increasing domain size and spacing reduced the interfacial contact area between soft and hard domains and thus

reduced the material mechanical properties. Finally, a highly detailed and geometrically complex object was successfully 3D printed, underscoring the ability of the present technique to fabricate customized objects using digital-to-digital technology. The findings of this work will facilitate the fabrication of 3D printed structures with precise nanoscopic features and bulk properties not accessible through traditional synthetic approaches. The ability to 3D print nanostructured materials in arbitrary shapes opens new avenues for advanced materials manufacturing, in which functional domains may be directly integrated within 3D printed objects. Considering the versatility of PIMS approach and the capabilities of 3D printing, a wide variety of 3D printable nanostructured materials can be produced for diverse applications including energy storage, catalysis, and drug delivery.

## Methods

**Synthesis and characterization of polymers.** Details of the syntheses including materials used, and complete polymer characterization by SEC and proton nuclear magnetic resonance spectroscopy ($^1$H NMR) are provided in the Supplementary Synthetic Procedures section.

**Polymerization kinetics study of resins.** For the kinetics study, resins with different formulations were prepared according to Supplementary Table 2. It should be noted that the amount of inhibitor contained in AA and PEGDA was compensated by the addition of extra TPO in a 1:1 molar ratio. The reaction mixture was then covered in aluminum foil and vortexed for 20 s prior to irradiation using the protocol described in the section "Attenuated total reflectance—Fourier transform infrared (ATR-FTIR) spectroscopy" of the Supplementary Information.

**3D printing setup and procedure.** A typical procedure for fabricating 3D printed objects is as follows: A 3D object was designed using Tinkercad 3D modeling software and the object was exported as a stereolithography (STL) file. The STL file was opened using Photon Workshop where the Z lift speed was set to 3 mm/s and the Z retract speed was set to 2 mm/s, while the Z lift distance was set to 6 mm. Printing parameters, such as layer thickness and exposure time, were defined in Photon workshop, then the model was sliced and exported as a photon workshop slice (PWS) file for 3D printing. The PWS file was copied to a flash drive for use with a masked DLP 3D printer (Anycubic Photon S) with a violet ($\lambda_{max} = 405$ nm) light LED array ($I_0 = 0.4$ mW cm$^{-2}$, as measured at the digital mask surface using a Newport 843-R power meter). For 3D printed samples for DMA and tensile tests, the layer thickness was 100 μm, off time was 2 s, layer and bottom exposure times were 180 s, number of bottom layers was 2. Typical 3D printing resin formulations were prepared by combining the calculated amounts of PBA$_n$-CTA, AA, PEGDA, and TPO (Supplementary Table 2). It should be noted that the amount of inhibitor contained in AA and PEGDA was compensated by the addition of extra TPO in a 1:1 molar ratio. The resin was then added to the 3D printer vat, and the desired print program was run. After 3D printing was completed, the printed objects were separated from the build plate, washed with ethanol, air dried, and post-cured under violet light ($\lambda_{max} = 405$ nm) for 15 min.

The cubic lattice model was designed (Supplementary Fig. 28) and 3D printed with a cure time per layer of 25 s/layer and a layer thickness of 100 μm using an Anycubic Photon Mono X 3D printer (light-source: high-quality filament ($\lambda_{max} = 405$ nm, $I_0 = 0.9$ mWcm$^{-2}$)). The cubic lattice had an overall length, width, and thickness of 30 mm and a strut width of 0.9 mm. The lattice model for compression testing was designed (Supplementary Fig. 31) and 3D printed with a layer thickness of 50 μm and cure times per layer of 25 s/layer and 52 s/layer for PIMS and non-PIMS resins, respectively, using an Anycubic Photon Mono X 3D printer (light-source: high-quality filament ($\lambda_{max} = 405$ nm, $I_0 = 0.9$ mWcm$^{-2}$)).

**Characterization of 3D printed materials.** The morphology and mechanical properties of 3D printed materials were characterized by AFM, SAXS, DMA, SEM, tensile, and compression testing. Details of material characterization using these methods are provided in the Supplementary Materials and Methods section.

## Data availability

The data generated in this study are provided within the article and its Supplementary Information file. Extra data are available from the corresponding author upon request. Source data are provided with this paper.

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

## Acknowledgements

J.Z., N.C., and C.B. acknowledge support by the Australian Research Council via Discovery Research program (DP210100094). The authors acknowledge the facilities and technical assistance provided by the Mark Wainwright Analytical Centre at UNSW Sydney for support in AFM and SEM imaging and Paul Fitzgerald and Sydney Analytical at The University of Sydney for support in SAXS experiments.

## Author contributions

V.A.B., J.Z., N.C., and C.B. conceived the research. V.A.B., N.C., and C.B. designed the experiments. V.A.B., X.S., and Y.X. carried out the experiments. Y.Y. performed AFM characterization. J.Z., N.C., and C.B. supervised the study. V.A.B., J.Z., N.C., and C.B. were involved in the data interpretation, discussed the results, and commented on the study. V.A.B., N.C., and C.B. wrote the manuscript with input from all authors. All authors reviewed and approved the manuscript.

## Competing interests

The authors declare no competing interests.
