## [Peer Review File · Nature Communications]

Nano- to macro-scale control of 3D printed materials via polymerization induced microphase separationReviewers' Comments:

Reviewer #1:

Remarks to the Author:

The authors provide an in-depth exploration of nanostructured materials fabricated by 3D printing using polymerization-induced microphase separation. The work is thorough however it does appear to be for the most part an expansion of their previous work in *Advanced Materials* (2022, 34, 2107643).

This work develops the previous work by using SAXS to gain insight into the structure of the soft and hard domains and correlate structural parameters to their mechanical properties. The results from this study are valuable and will provide insight for future work in this area.

My questions for the authors:

1. Have the authors measured viscosity as a function of PBA -CTA loading and/or length? It would be nice to have a sense of the magnitude of change in viscosities as these two parameters are varied. Viscosities could play a role the balance between the kinetics and thermodynamics of the phase separation. The viscosity differences could also explain in part why the resin systems with short PBA chains polymerize more slowly (more opportunity for termination reactions).
2. The authors should point out in the main body of the manuscript that the light intensity of the 3D printed parts is 0.4mW/cm² while that of the FTIR study is 2mW/cm². Do the authors anticipate that the conversion profiles different significantly under these two intensities?
3. The authors correlate D(PBA) and d(saxs) with some of the mechanical properties of the material- with an ideal domain size and domain spacing that enables better stress dissipation in a bicontinuous morphology. Could the authors expand on the idea of interfacial segregation (as quantified by the f factor) and how it could impact toughness?
4. The authors missed an opportunity to demonstrate the value of 3D printing these materials as lattices. Have the authors characterized any of the 3D lattices for the tensile or compression properties?

Reviewer #2:

Remarks to the Author:

The authors present a 3D printing method in which nanoscale structural features can be controlled. This is made possible through suitable macromolecular chain transfer reagents which can microphase separate. As the control over the macroCTA's structure controls the resulting microdomains, the properties of the 3D printed objects can be tuned in their mechanical performance. The interaction of soft and hard domains in the nanostructures are highly controllable and the method bridges polymerization induced microphase separation (PIMS) and 3D printing. The work shows for the first time that with a range of different macro-CTA's the development of structure-property relationships between the initial resin components and nanostructured morphologies is possible. Moreover, the bulk mechanical properties with an optimal length scale for a bicontinuous phase was observed. The 3 D printed materials were prepared in a robust approach with high versatility and probability to design hierarchically constructed materials.

The experiments are conducted with a deep understanding of macroCTA preparation, effect of macro-CTA chain length on polymerization kinetics and a detailed investigation of the morphology evolution in the 3D printed PIMS materials. Surface morphologies were investigated using the dependence of the macro-CTAs degree of polymerization and loading. Mechanical testing was conducted and is particularly notable for materials 3D printed with a lower loading of macro-CTA. It was attributed to the morphology transition from discrete globular nanodomains to more continuous, interpenetrating soft and hard nanodomains and allowing a more even distribution of stress. Furthermore, the capability of the RAFT-mediated PIMS 3D printing was demonstrated with the macroscale geometric control which is difficult to achieve by traditional manufacturing.

The strength of the presented work is the robustness of the 3D printing technique and the achieved

excellent nanostructural control of the material with advanced properties. This technique will accelerate the development of materials that will play an important role in our society such as energy storage materials .

Reviewer #3:

Remarks to the Author:

The manuscript by Bobrin et al. studies polymerization-induced phase separation in the context of 3D printing by digital light processing. Overall, the manuscript is detailed and convincing, but its scope is nearly identical to the authors' previous work just published in *Advanced Materials* (Bobrin, *Adv. Mater.* 2022, 10.1002/adma.202107643). The same process and materials are used (e.g., compare Figure 1 in both papers), including 2 out of the 5 macromonomer molecular weights ($X_n = 24$ and 48) reported here. Moreover, analogous polymerization-induced morphological transitions (discrete to continuous) and mechanical properties were observed in both papers as a function of formulation design. In this reviewer's opinion, the current manuscript is quite incremental over their previous work (which is great) and does not have the novelty that would justify publication in *Nature Communications*.

RESPONSE POINT-BY-POINT:

Reviewers' comments are in blue.

Author responses are in regular text.

Quotes from the manuscript are italicized

All new parts to the main text and supporting information are highlighted in the revised versions.

COMMENTS TO AUTHOR:

Reviewer #1: The authors provide an in-depth exploration of nanostructured materials fabricated by 3D printing using polymerization-induced microphase separation. The work is thorough however it does appear to be for the most part an expansion of their previous work in *Advanced Materials* (2022, 34, 2107643).

This work develops the previous work by using SAXS to gain insight into the structure of the soft and hard domains and correlate structural parameters to their mechanical properties. The results from this study are valuable and will provide insight for future work in this area.

Response: We thank the reviewer for their positive and insightful comments. Please find a point-by-point response to the reviewers' comments below.

Although this present work appears to have some similarity with our previous work (*Adv. Mater.* 2022, 10.1002/adma.202107643), we feel that the current work provides novel and significant data and represents an important contribution to field of 3D printing of nanostructured materials. Compared to our previous work, where we demonstrated a proof-of-concept bridging PIMS and 3D printing to fabricate nanostructured materials, the current work focuses on developing a fundamental understanding of structure-property relationships for nanostructured 3D printed PIMS materials. In our previous publication, limited control over material nanostructuration was reported via blending non-functional homopolymers and a macroCTA with a fixed chain length. Importantly, we were not able to control the domain spacing and size by this blending approach. To illustrate the limitations of our previous study, we have updated **Figure 1**.

In the present work, we demonstrated the fabrication of 3D printed materials with highly tunable morphologies and precisely controllable domain sizes and domain spacings. This was achieved by establishing the relationship between the macroCTA degree of polymerization and the macroCTA concentration in determining the ultimate morphology produced. Importantly, the domain sizes of these nanostructured materials followed predictable scaling behavior, increasing according to a power law with increasing block copolymer size. As such, the nanostructures of these 3D printed materials can be

predetermined. This is essential in designing 3D printed PIMS materials with predictable structure-property relationships as we demonstrated that bulk mechanical properties of 3D printed PIMS materials are largely defined by type of morphology and domain length scale, e.g., there is an optimum length scale to obtain the highest mechanical properties for 3D printed materials with interpenetrating morphologies, i.e., elongated domains and bicontinuous morphology, which has never been reported before for PIMS materials. Furthermore, newly performed compression tests demonstrate that the materials prepared by PIMS display considerably increased mechanical properties compared to their non-PIMS counterparts (see Supplementary Fig. 29 and Supplementary Table 8). Detailed investigation of the internal nanostructures via SAXS using the Teubner-Stray (T-S) model was also performed, which demonstrated that the segregation strength at the interfaces (sharpness at the interfaces) is slightly affected by the macroCTA degree of polymerization, which in turn influences the material mechanical properties. Altogether, this work opens new opportunities in the design of functional 3D printed materials with predefined morphology and mechanical performance.

1. Have the authors measured viscosity as a function of PBA -CTA loading and/or length? It would be nice to have a sense of the magnitude of change in viscosities as these two parameters are varied. Viscosities could play a role the balance between the kinetics and thermodynamics of the phase separation. The viscosity differences could also explain in part why the resin systems with short PBA chains polymerize more slowly (more opportunity for termination reactions).

Response: We agree with the reviewer's comment. We measured the viscosities of the selected resins to investigate the effect of PBA-CTA X_n and PBA-CTA wt.%. Please see the new Supplementary Table 3 in the Supplementary Information.

"The viscosities of the selected resins were determined to investigate the effects of PBA-CTA X_n and PBA-CTA wt%. At fixed wt% of PBA-CTA (16.5 wt%), the viscosities of resins increase from 9.97 mPa s for PBA₂₄-CTA to 40.65 mPa s for PBA₃₆₀-CTA. At fixed PBA-CTA X_n ($X_n = 24$), the viscosities of resins increase from 9.97 mPa s at 16.5 wt% loading to 17.65 mPa s at 28.2 wt% loading and further to 46.41 mPa s at 43.9 wt% loading. The similar trend was observed for the viscosities of resins consisted of PBA₃₆₀-CTA as a function of wt% (40.65 mPa s at 16.5 wt%, 123.6 mPa s at 28.2 wt% and 494.1 mPa s at 43.9 wt%). It should be noted that all resins were suitable for DLP 3D printing".

In line with these changes, we added the sentence in the main text discussion.

Page 5

"The viscosity of resins increases upon increasing PBA-CTA X_n and wt% of PBA-CTA (Supplementary Table 3)".

We agree with the reviewer's comment that viscosity could play a role in the balance between the kinetics and thermodynamics of the phase separation. For a fixed wt% of PBA-CTA, there is a clear trend between the viscosity of the resin and the f_a values (the sharpness at the interfaces). For instance, upon increasing the resin viscosity from 9.97 mPa s ($X_n = 24$) to 40.65 mPa s ($X_n = 360$) the f_a values of PIMS materials increase from -0.91 ($X_n = 24$) to -0.77 ($X_n = 360$), indicating the formation of more diffuse interface (Supplementary Table 3 and Supplementary Table 6). This could be due to reduced polymer chain mobility upon increasing viscosity, which could lead to the formation of domains with more diffuse interface. However, by comparing two resins consisted of the same macroCTA of PBA₃₆₀-CTA (16.5 wt% loading), but with different TPO wt%, 0.3 wt% and 0.133 wt%, respectively, we observed different f_a values, -0.77 and -0.68, respectively, though the viscosity of resins is the same (40.65 mPa s) (Supplementary Table 3 and Supplementary Table 6). Based on these results, we can conclude that there is the complex interplay among factors, e.g., resin viscosity, molar ratio between macroCTA and TPO, and macroCTA X_n and loading, that influence microphase separation.

Regarding the effect of viscosity on polymerization kinetics, we agree with the reviewer that the viscosity could explain in part the difference in polymerization kinetics. For 16.5 wt% loading of PBA-CTA, the resins demonstrated faster polymerization kinetics with increasing the viscosity from 9.97 mPa s ($X_n = 24$) to 40.65 mPa s ($X_n = 360$) (Fig. 2a and Supplementary Table 3), which can be explained by the Trommsdorff–Norrish effect (Nature, 1942, 150, 336–337; Makromol. Chem., 1948, 1, 169–198).

We have updated discussion in the main text to reflect this point, which now reads as:

Page 6

“The slight reduction in polymerization rate for resins with low macroCTA chain lengths ($X_n = 24$ and 48) can be explained by lower resin viscosity and the increased concentration of RAFT end-groups, in accordance with previous observations”.

However, we should note that concentration ratio of macroCTA to TPO also plays a significant role in polymerization kinetics. Indeed, by comparing two resins consisted of the same macroCTA of PBA₃₆₀-CTA (16.5 wt% loading), but with different TPO wt%, 0.3 wt% and 0.133 wt%, respectively, we observed different conversion profiles, though the viscosity of resins is the same (40.65 mPa s) (Supplementary Table 3 and Supplementary Fig. 24c).

2. The authors should point out in the main body of the manuscript that the light intensity of the 3D printed parts is 0.4mW/cm² while that of the FTIR study is 2mW/cm². Do the authors anticipate that the conversion profiles different significantly under these two intensities?

Response: We agree with the reviewer's comment. We updated the main text to point out the difference in the light intensity between FTIR resin kinetics study and 3D printing. Please see below:

Page 5

“The photopolymerization kinetics of each resin was then determined in open-air conditions under 2.06 mW cm⁻² violet light irradiation ($\lambda_{max} = 405$ nm)”.

Page 7

“To investigate the effect of PBA-CTA X_n and wt% on the nanostructure of 3D printed materials, the 15 resins were applied to a commercial DLP 3D printer ($I_0 = 0.4$ mW cm⁻², $\lambda_{max} = 405$ nm) to fabricate model rectangular prisms ($l \times w \times t = 40 \times 8 \times 2$ mm)”.

In general, we use FTIR kinetics study to investigate the effect of concentration of resin components on polymerization kinetics (e.g., TPO concentration, RAFT agent concentration) and therefore optimize the resins before 3D printing. As demonstrated in our previous works (Angew. Chem. Int. Ed. 2019, 58, 17954–17963; Angew. Chem. 2021, 133, 8921 – 8932), there is good correlation between conversion profiles of resins obtained in FTIR study and 3D printing parameters setup (e.g., layer cure time), suggesting that conversion profiles do not vary significantly despite the difference in light intensity. In addition, there is another literature reference (Guymon et al., Additive Manufacturing 27 (2019) 20–31), which indicates that overall trend of the reaction rates determined by FTIR remains consistent with DLP 3D printing conditions. In this work, FTIR kinetics study of resins was performed under 2.06 mW cm⁻² violet light irradiation ($\lambda_{max} = 405$ nm) using a small quantity of resin placed on the top of ATR-FTIR instrument. This intensity value is the lowest that we can use in our FTIR setup. The results of FTIR kinetics study were used to identify the trends in resin double bond conversion profiles depending on PBA-CTA X_n .

3. The authors correlate $D(\text{PBA})$ and $d(\text{saxs})$ with some of the mechanical properties of the material- with an ideal domain size and domain spacing that enables better stress dissipation in a bicontinuous morphology. Could the authors expand on the idea of interfacial segregation (as quantified by the f factor) and how it could impact toughness?

Response: f_a parameters were determined from fitting SAXS data using the Teubner-Stray (T-S) model. The f_a represents the segregation strength at the interfaces (sharpness at the interfaces). To elucidate the possible effect of f_a on toughness, we compared two PIMS materials 3D printed using the same macroCTA of PBA₃₆₀-CTA at 16.5 wt% loading, but with different TPO wt%, 0.3 wt% and 0.133 wt%. These two materials have identical structures: both materials feature bicontinuous morphologies with very similar values of PBA domain width (D_{PBA}) and domain spacing (d_{AFM}): D_{PBA} and $d_{\text{AFM}} = 23 \pm 2$ nm and 55 ± 7 nm

for PIMS material 3D printed with PBA₃₆₀-CTA with 0.3 wt% TPO, and D_{PBA} and $d_{AFM} = 29 \pm 6$ nm and 55 ± 7 nm for PIMS material 3D printed with PBA₃₆₀-CTA with 0.133 wt% TPO. Fitting the SAXS data using the T-S model revealed different f_a values: $f_a = -0.77$ for the PIMS material 3D printed with PBA₃₆₀-CTA with 0.3 wt% TPO, while for material 3D printed with 0.133 wt% TPO $f_a = -0.68$, indicating the formation of domains with more diffuse interface, i.e., higher extent of mixing between PBA and *net*-P(AA-*co*-PEGDA) phases. Tensile tests of 3D printed dog-bone specimens revealed that 3D printed materials with lower f_a value ($f_a = -0.77$) exhibited improved mechanical properties compared to material with larger f_a value ($f_a = -0.68$): tensile stress at break (σ_B) increased from 39.2 to 47 MPa, elongation at break (ϵ_B) increases from 63% to 82.2%, and toughness increases from 20.3 to 31.5 MJ m⁻³. These results suggest that 3D printed PIMS materials with sharper interface (lower f_a values) exhibit improved mechanical properties.

In line with these changes, new Supplementary Fig. 24 and Supplementary Table 7 have also been included in the Supplementary Information showing the AFM images, the corresponding domain spacing histograms and mechanical properties. Furthermore, we have included further discussion in the main text, which now reads as:

Page 16

*“It is worth noting that the sharpness of interphase between PBA and net-P(AA-*stat*-PEGDA) domains, as quantified by f_a parameter, may impact the toughness of 3D printed PIMS materials. To investigate this, two PIMS materials 3D printed using the same macroCTA of PBA₃₆₀-CTA at 16.5 wt% loading, but with different TPO wt%, 0.3 and 0.133 wt%, were compared. Both materials feature bicontinuous morphologies with very similar values of D_{PBA} and d_{SAXS} (Supplementary Fig. 24). Fitting SAXS data using the T-S model revealed different f_a values: $f_a = -0.77$ and -0.68 for the materials 3D printed with 0.3 and 0.133 wt% TPO, respectively. This result suggests the formation of domains with more diffuse interfaces for materials 3D printed with lower TPO wt%, which may result from the difference in the polymerization rate (Supplementary Fig. 24). Consequently, the material 3D printed with higher TPO wt% exhibited improved mechanical properties: σ_B and ϵ_B increased from 39.2 to 47 MPa and from 63% to 82.2%, respectively, resulting in the increase in toughness from 20.3 to 31.5 MJ m⁻³ (Supplementary Table 7)”.*

4. The authors missed an opportunity to demonstrate the value of 3D printing these materials as lattices. Have the authors characterized any of the 3D lattices for the tensile or compression properties?

Response: We agree that the inclusion of compression properties of 3D printed lattices can be beneficial in the current paper. As such, body-centered cubic lattice structures were 3D printed using PIMS and non-PIMS resins to demonstrate the effect of nanostructuration on compression properties of 3D printed lattices. Notably, PIMS materials exhibited considerably increased mechanical properties compared to non-PIMS

materials. For instance, PIMS lattice structures demonstrated a 300% increase in Young's modulus, ~920% increase in modulus of resilience, and a 233% increase in toughness compared to non-PIMS analogues (non-phase separated). In line with these changes, new Supplementary Figs. 28-29 and new Supplementary Table 8 have been included. These new figures and tables show photos of 3D printed lattice structures, the compressive stress-strain curves, and the lattice mechanical properties. Furthermore, we have included further discussion in the main text, which now reads as:

Page 18

“Furthermore, to demonstrate the impact of material nanostructuring on mechanical properties of 3D printed lattices, resins with and without PBA-CTA were selected to 3D print body-centered cubic lattice structures (Supplementary Fig. 28). Uniaxial compression testing was conducted to obtain the stress-strain curves of PIMS and non-PIMS lattice structures (Supplementary Fig. 29). Notably, the mechanical properties of materials 3D printed using PIMS resins were significantly higher than non-PIMS materials (Supplementary Table 8). Specifically, the Young's modulus, modulus of resilience, and toughness increased by 4-, 10-, and 3-fold, respectively, compared to non-PIMS counterparts. These results highlight the beneficial effect of nanostructuring to enhance mechanical properties of 3D printed materials”.

Reviewer #2: The authors present a 3D printing method in which nanoscale structural features can be controlled. This is made possible through suitable macromolecular chain transfer reagents which can microphase separate. As the control over the macroCTA's structure controls the resulting microdomains, the properties of the 3D printed objects can be tuned in their mechanical performance. The interaction of soft and hard domains in the nanostructures are highly controllable and the method bridges polymerization induced microphase separation (PIMS) and 3D printing. The work shows for the first time that with a range of different macro-CTA's the development of structure-property relationships between the initial resin components and nanostructured morphologies is possible. Moreover, the bulk mechanical properties with an optimal length scale for a bicontinuous phase was observed. The 3D printed materials were prepared in a robust approach with high versatility and probability to design hierarchically constructed materials.

The experiments are conducted with a deep understanding of macroCTA preparation, effect of macro-CTA chain length on polymerization kinetics and a detailed investigation of the morphology evolution in the 3D printed PIMS materials. Surface morphologies were investigated using the dependence of the macro-CTAs degree of polymerization and loading. Mechanical testing was conducted and is particularly notable for materials 3D printed with a lower loading of macro-CTA. It was attributed to the morphology transition from discrete globular nanodomains to more continuous, interpenetrating soft and hard nanodomains and allowing a more even distribution of stress. Furthermore, the capability of the RAFT-mediated PIMS 3D printing was demonstrated with the macroscale geometric control which is difficult to achieve by traditional manufacturing.

The strength of the presented work is the robustness of the 3D printing technique and the achieved excellent nanostructural control of the material with advanced properties. This technique will accelerate the development of materials that are will play an important role in our society such as energy storage materials.

Response: We would like to thank the reviewer for their positive comments.

Reviewer #3: The manuscript by Bobrin et al. studies polymerization-induced phase separation in the context of 3D printing by digital light processing. Overall, the manuscript is detailed and convincing, but its scope is nearly identical to the authors' previous work just published in *Advanced Materials* (Bobrin, *Adv. Mater.* 2022, 10.1002/adma.202107643). The same process and materials are used (e.g., compare Figure 1 in both papers), including 2 out of the 5 macromonomer molecular weights ($X_n = 24$ and 48) reported here. Moreover, analogous polymerization-induced morphological transitions (discrete to continuous) and mechanical properties were observed in both papers as a function of formulation design. In this reviewer's opinion, the current manuscript is quite incremental over their previous work (which is great) and does not have the novelty that would justify publication in *Nature Communications*.

Response: We thank the reviewer for their insightful comments. We feel that the current work provides novel and significant data and represents an important contribution to field of 3D printing of nanostructured materials. Compared to our previous work published in *Advanced Materials* (*Adv. Mater.* 2022, 10.1002/adma.202107643), where we demonstrated a proof-of-concept bridging PIMS and 3D printing to fabricate nanostructured materials, this work focuses on fundamental understanding of structure-property relationships for the nanostructured 3D printed PIMS materials.

In our previous publication, we used a blending approach where non-functional homopolymer and a macroCTA with a fixed chain length were blended. We observed limited control over material nanostructuration and lack of tunability in domain spacing. In the present work, we demonstrated the fabrication of 3D printed materials with highly tunable morphologies with precise tuning of domain size and domain spacing. Importantly, the domain sizes of these nanostructured materials followed predictable scaling behavior, increasing according to a power law with increasing block copolymer size. As such, the nanostructures of these 3D printed materials can be predetermined. This is essential in designing 3D printed PIMS materials with predictable structure-property relationships. Indeed, we demonstrated that bulk mechanical properties of 3D printed PIMS materials are largely defined by type of morphology and domain length scale. We uncovered an optimum length scale to obtain the highest mechanical properties for 3D printed materials with interpenetrating morphologies i.e., elongated domains and bicontinuous morphologies, which has never been reported before for PIMS materials. Furthermore, we have investigated for the first time the compression properties of these 3D printed PIMS materials and compare versus non-PIMS materials, i.e., materials without nanostructuration but formed using analogous monomer composition. We observed a significant enhancement in mechanical properties for 3D printed PIMS materials versus 3D printed non-PIMS materials. For instance, PIMS lattice structures demonstrated a 300% increase in Young's modulus, ~920% increase in modulus of resilience, and a 233% increase in toughness compared to non-PIMS analogues (non-phase separated) (See Supplementary Figs. 28-29 and Supplementary Table 8).

Furthermore, the current work clearly outlines the morphological evolution in PIMS materials. For the first time in PIMS literature, we provide a thorough breakdown of the co-dependence of the macro-CTA degree of polymerization and the macro-CTA concentration in determining the ultimate morphology produced. Previously, it has been shown that a randomly distributed spherical morphology can exist at low concentrations of macro-CTA; indeed, other works (and our previous work) point towards the macro-CTA concentration as being the most influential parameter on morphological evolution. However, this work clearly demonstrates that the morphology is both dependent on the macro-CTA molecular weight and the concentration, e.g., using 16.5 wt% PBA₉₄-CTA provides materials with elongated globular domains, while using 16.5 wt% PBA₁₈₀-CTA provides materials with bicontinuous morphologies; using 43.9 wt% PBA₉₄-CTA also provides materials with bicontinuous morphologies. When combined with the power law dependence on the domain spacing, this work thus provides comprehensive and systematic design rules for nanostructured PIMS materials. We have updated the introduction and **Figure 1** to reflect the novelty of our work more clearly and show the difference between the current work and our previous work (Adv. Mater. 2022, 10.1002/adma.202107643).

Other corrections from Authors:

The caption of **Figure 1** was updated. The caption now reads as: “*3D printing materials by RAFT-mediated PIMS. (a) PIMS mechanism: a macroCTA is chain extended with acrylic acid (AA) and poly(ethylene glycol) diacrylate (PEGDA) to form block copolymers, which eventually phase-separate with the generation of emergent morphologies trapped by in-situ cross-linking. (b) Previous work using PBA-CTA with fixed chain length blended with non-functional PBA to 3D print PIMS materials with limited control over morphology, and (c) current work using PBA-CTAs with varied chain length to investigate structure-property relationships of 3D printed PIMS materials*”.

Methods description, specifically the section “3D printing setup and procedure” was updated to add the details of 3D printing of BC lattice structures. The added part reads as: “*The lattice model was designed (Supplementary Fig. 28) and 3D printed with a layer thickness of 50 μm and the cure time per layer of 25 s/layer and 52 s/layer for PIMS and non-PIMS resins, respectively, using Anycubic Photon Mono X 3D printer (light-source: high-quality filament ($\lambda_{\text{max}} = 405 \text{ nm}$, $I_0 = 0.9 \text{ mWcm}^{-2}$))*”.

REVIEWERS' COMMENTS

Reviewer #2 (Remarks to the Author):

The authors have addressed all the comments from the reviewers. It ready to get accepted.

Reviewer #3 (Remarks to the Author):

I stand by my original review: this is a very incremental advance over the authors' just published paper in Advanced Materials. The current paper is publishable, but it is not a fundamentally new concept or major advance. Consequently, in this reviewer's opinion, it is more suitable for a specialized polymer journal like Polymer Chemistry or Macromolecules.

RESPONSE POINT-BY-POINT:

Reviewers' comments are in blue.

Author responses are in regular text.

Reviewer #2 (Remarks to the Author):

The authors have addressed all the comments from the reviewers. It ready to get accepted.

Response: We thank the reviewer for their positive support.

Reviewer #3 (Remarks to the Author):

I stand by my original review: this is a very incremental advance over the authors' just published paper in *Advanced Materials*. The current paper is publishable, but it is not a fundamentally new concept or major advance. Consequently, in this reviewer's opinion, it is more suitable for a specialized polymer journal like *Polymer Chemistry* or *Macromolecules*.

Response: We thank the reviewer for their comments, and we respect their opinion. However, we would like to emphasize the novelty of this paper, which was highlighted by the incorporation of new Figure 1 during our first revision. Although in our previous work (*Adv. Mater.* 2022, 10.1002/adma.202107643), we demonstrated some control of the nanostructuration during 3D printing using a blending approach where non-functional homopolymer and a macroCTA with a fixed chain length were blended. We observed limited control over material nanostructuration and lack of tunability in domain spacing. In the present work, we overcame these deficiencies and demonstrated the fabrication of 3D printed materials with highly tunable morphologies (globular to bicontinuous morphology) with precise tuning of domain size and domain spacing. This is the first example in the literature which shows such control in 3D printing using a vat photopolymerization. Importantly, the domain sizes of these nanostructured materials followed predictable scaling behavior, increasing according to a power law with increasing block copolymer size. As such, the nanostructures of these 3D printed materials can be predetermined. This is essential in designing 3D printed PIMS materials with predictable structure-property relationships. Furthermore, the current work clearly outlines the morphological evolution in PIMS materials. For the first time in PIMS literature, we provide a thorough breakdown of the co-dependence of the macro-CTA degree of polymerization and the macro-CTA concentration in determining the ultimate morphology produced. Finally, we established that the nanostructuration of these 3D printing objects can enhance their mechanical performance in comparison with non-nanostructured materials with analogous composition. Therefore, we believe that the design paradigms developed in this work will enable the fabrication of hierarchically structured materials with medical,

engineering, and energy applications. For all these reasons, we believe that this manuscript is suitable for *Nature Communications*.